# Exploring the Nexus of Climate Change and Substance Abuse: A Scoping Review

**DOI:** 10.3390/ijerph21070896

**Published:** 2024-07-09

**Authors:** Luca Tomassini, Massimo Lancia, Angela Gambelunghe, Abdellah Zahar, Niccolò Pini, Cristiana Gambelunghe

**Affiliations:** 1School of Advanced Studies, University of Camerino, 62032 Camerino, Italy; luca.tomassini@unicam.it; 2Forensic Medicine, Forensic Science and Sports Medicine Section, Department of Medicine and Surgery, University of Perugia, 06129 Perugia, Italy; abdellah.zahar@studenti.unipg.it (A.Z.); niccolo.pini@studenti.unipg.it (N.P.); cristiana.gambelunghe@unipg.it (C.G.); 3Occupational Medicine, Respiratory Diseases and Toxicology Section, Department of Medicine and Surgery, University of Perugia, 06129 Perugia, Italy; angela.gambelunghe@unipg.it

**Keywords:** climate change, substance abuse, drugs, heatwaves, health risks

## Abstract

Introduction: The increase in average air temperature and multiple extreme weather events, such as heatwaves and droughts, pose significant health risks to humans. This scoping review aims to examine the current state of the existing literature concerning the potential relationship between substance abuse and climate change, along with the aspects it encompasses. Material and methods: The review followed PRISMA guidelines for methodological rigor, aiming to identify studies on drug abuse. Searches were conducted across the primary databases using specific search strings. Quality assessment involved evaluating the research question’s clarity, search strategy transparency, consistency in applying the inclusion/exclusion criteria, and reliability of data extraction. Results: Most studies were conducted in the USA. They included observational and retrospective quantitative studies, as well as qualitative and prospective observational ones. Research examined the correlation between extreme weather and some substance abuse. All studies analyzed the adverse effects of climate change, especially heatwaves, on both physiological and pathological levels. Conclusions: The scoping review notes the scarcity of studies about the correlation between substance abuse and climate change, and emphasizes the threats faced by individuals with substance abuse and mental health disorders due to climate change.

## 1. Introduction

Climate change has been extensively documented as a critical factor impacting human health, posing significant risks through direct consequences such as heatwaves and changes in air quality, as well as indirect effects altering ecosystems and socio-economic systems globally [1,2,3].

According to several studies, the average air temperature above the Earth’s surface has increased by 1.53 °C compared to the pre-industrial era, which spans from 1850 to 1900 [4,5].

The World Meteorological Organization (WMO) reported a 66% likelihood that the global temperature will exceed 1.5 °C above pre-industrial levels for at least one year between 2023 and 2027 [6]. Furthermore, NASA’s Climate Change site indicates that, as of 2023, both land and ocean surface temperatures have risen significantly, with land temperatures increasing by 2.10 °C and ocean temperatures by 1.10 °C compared to the 1850–1900 average [3].

The complexity of extreme-event impacts is exacerbated by the interaction of multiple extremes, referred to as compound events, which occur either simultaneously or sequentially across different times and locations [7,8,9]. These compound events, such as consecutive heatwaves or heatwaves coinciding with droughts, pose significant risks to human health [10].

However, research in this area is still evolving, and our understanding of current and future risks, along with the societal factors influencing vulnerability, remains limited. For instance, the amplification of heatwaves by air pollution creates hazardous health conditions [4,11,12].

Research is increasingly uncovering connections between climate change and various health outcomes, including mental health effects, highlighting its multifaceted impact on human activities and public health [13,14,15,16].

During extreme weather conditions, such as during severe heatwaves, the demand for medical emergency services increases, leading to management challenges [15,17].

In the realm of substance abuse, despite the literature being somewhat limited, a correlation is suggested between substance use disorder and climatic phenomena, situated within a wider discourse on the association between climate change and the incidence of psychiatric disorders [18]. Additionally, the potential for divergent individual responses to substance use during periods of extreme climatic conditions has been contemplated [15,18,19]. The issues related to substance abuse and climate change, seemingly distinct, are increasingly recognized as interconnected and necessitate a comprehensive understanding and further research to address their complex interaction [20].

For instance, some studies have highlighted that substance users, especially those utilizing stimulants such as cocaine and amphetamines, encounter heightened health risks during extreme heat conditions owing to their altered thermoregulation, rendering them more susceptible to temperature fluctuations [21,22,23,24]; similar effects have also been highlighted for alcohol and opioids [25].

Therefore, it appears that climate change, with its rising temperatures, extreme weather events, and environmental degradation, poses a significant threat to human health, well-being, and drug abuse [18,20].

This review aims to examine the biomedical literature to understand the connections between substance abuse and climate change, with particular attention being paid to the epidemiological, pathophysiological, and toxicological aspects. It is important to emphasize that climate change is a complex and multifaceted global phenomenon that can be analyzed from various perspectives, primarily sociological [26,27]. The events under scrutiny encompass a range of global issues: the global financial crisis, economic sanctions, political transitions affecting ethnic minorities, colonialism’s impact on indigenous communities, and ecological disasters. These events have inflicted trauma, displacement, and severe disruptions in essential healthcare services for people who use drugs, leaving them significantly underserved during these crises [28,29].

With this clarification, this scoping review aims to delve into the connections between climate change and substance abuse by examining the existing literature. Specifically, this review is geared towards evaluating the current state of research on this topic, encompassing the primary themes explored, the issues under discussion, and the range of substances involved in abuse.

## 2. Materials and Methods

This review adhered to the Preferred Reporting Items for Systematic Reviews and Meta-Analyses (PRISMA) guidelines, ensuring methodological rigor and transparency throughout the process. The aim was to identify research studies exploring the intersection of substance abuse and climate change within the timeframe of 1 January 2018 to 31 December 2023.

A systematic search was conducted across three electronic databases, including PubMed, Web of Science, and Scopus. The search strategy was adapted from a PubMed search string (A), customized for each specific database to ensure comprehensive coverage (B). While recognizing the broader construct of climate change, the search terms were intentionally delimited to focus on specific aspects, such as drug abuse and substance abuse, to manage the volume of retrieved references effectively, which were in the thousands.

Title/abstract and full-text screening were independently conducted by three independent researchers using Rayyan AI software.

The conflicts were resolved through discussion and consensus sought among the authors throughout this screening and extraction process to ensure accuracy and reliability. Data extraction from eligible studies was carried out independently by the three authors utilising a standardized data extraction form PDF file.

Firstly, a specific string adapted for each website was used, including mesh-terms and all fields:

((“Climate Change”[MeSH Terms] OR “Global Warming”[MeSH Terms] OR “Climate Crisis”[MeSH Terms] OR “Heat-Waves”[MeSH Terms] OR “Weather Modification”[MeSH Terms] OR “Climatic Disruption”[MeSH Terms] OR “Change in Earth’s Climate”[MeSH Terms] OR “Change in Global Climate”[MeSH Terms] OR “Climate Change Crisis”[MeSH Terms] OR “Global Climate Change” OR “Climate Change” OR “Climate Changes” OR “Global Warming” OR “Climate Crisis” OR “Heat-Waves” OR “Weather Modification” OR “Climatic Disruption” OR “Change in Earth’s Climate” OR “Change in Global Climate” OR “Climate Change Crisis”) AND (“Substance-Related Disorders”[MeSH Terms] OR “Drug Users”[MeSH Terms] OR “Recreational Drug Use”[MeSH Terms] OR “Substance Abuse, Intravenous”[MeSH Terms] OR “Illicit Drugs”[MeSH Terms] OR “Drug Addiction”[MeSH Terms] OR “Psychoactive Substances”[MeSH Terms] OR “Recreational Drugs” OR “Drugs” OR “Recreative Drug” OR “Recreative Drugs” OR “Substances” OR “Stupefy” OR “Narcotic” OR “Narcotics” OR “Illicit Drugs” OR “Drug Addiction” OR “Recreational Drug” OR “Recreational Drugs”))

Regarding the search string, it should be noted that outdated terms such as “drug users” were also employed to ensure a broader and more sensitive search. However, it is important to clarify that this is a stigmatizing term, and the current preferred terminology is “people who use drugs.”

The *inclusion criteria* for this review were as follows:-Publication must relate to substance abuse and climate change across one or more of the domains: climate change impacts, drug exposures, and vulnerability;-Publication must be original research;-Publication must be between the timeline of January 2018 to December 2023;-Papers that are about the epidemiology of recreational drugs and their diffusion in connection to the climate change were included;-Papers that are about climate change and substance abuse were included;-All kinds of recreational drugs were considered;-Articles taken from the citations of the studies examined that were not bound by the timeline of 5 years were considered.

The *exclusion* parameters established for this scoping review are outlined as follows:
-Systematic review, scoping review, conference presentation, and textbooks;-Any articles that do not encompass the topic of climate change;-Articles without a clear linkage between drug abuse and climate change;-If the topic is taking into consideration any medical treatment;-Studies conducted on psychiatric patients;-If the research is related to psychiatric disorder, including substance abuse.

Regarding the exclusion criteria, it is clarified that the aim of this scoping review is to explore the relationship between substance abuse and climate change, focusing specifically on contexts outside of psychiatry. The psychiatric context, being a distinct area of study, falls outside the scope of interest for this investigation.

Additionally, in the citation search, the emphasis was on identifying the most relevant and recent publications to understand the current state of the scientific literature on the topic. This difference in temporal approach aimed to balance historical understanding with the latest available evidence, thereby contributing to a comprehensive and updated overview of the field of study.

The process of selecting the works was carried out using the Rayyan software.

Rayyan is a free web and mobile application that assists in expediting the initial screening of abstracts and titles through a semi-automated process. It incorporates a high level of usability and can be accessed at the following website: http://rayyan.qcri.org [30,31].

### Quality Assessment

Assessing the quality of our scoping review was carried out by the same three authors and it involved evaluating several key components to ensure reliability.

The research question was broad enough and, at the same time, specific enough to guide the review process, by using a specific string and MeshTerm.

The search strategy was rendered comprehensive and transparent by utilizing public databases (PubMed, Scopus, and Web of Science) and restricting the timeline to five years. Additionally, several limitations were applied, including the inclusion criterion of English language papers and the exclusion of domains such as geology, sociology, and politics.

The same three authors also clearly defined together the inclusion and exclusion criteria and applied them consistently, and, to ensure reliability, the three authors conducted the data extraction using the AI Rayyan, which helped identify key words and themes.

## 3. Results

A total of 58,505 articles were initially identified and processed, comprising 1344 from PubMed, 55,043 from Scopus, and 2119 from Web of Science, in addition to those extracted from their bibliographies. Subsequently, 19,147 duplicate articles were removed. The screening of the remaining 39,358 articles was conducted based on the inclusion and exclusion criteria, resulting in 58 articles. Further screening excluded 11 articles lacking climate change information, 25 lacking consideration of substance abuse, 1 duplicate, and 11 articles referring to other specific subjects such as geology and politics. Consequently, a total of 10 articles were obtained. From the 50 articles extracted from bibliographies, 47 were excluded: 20 contained solely climate change information, 20 lacked any association with substance abuse, and 7 met one or more exclusion criteria. Thus, a total of 13 articles were included in the present scoping review, as described in Figure 1.

Regarding the country of the included studies, nine (69.2%) were conducted in USA, of which three were in California (37.5%), three in NY (37.5%), one in Connecticut (12.5%), two non-specific states in the USA (25%); one (7.69%) in Germany, one (7.69%) in the UK, one (7.69%) in Australia, and one (7.69%) in Canada.

In total, eight were observation quantitative studies, two were retrospective quantitative studies, one was a qualitive study, and one was a prospective observational study.

Six studies analyzed the correlation between extreme weather conditions and amphetamine, cocaine, opioids, and cannabis. In four studies, the effects of alcohol abuse and climate change were indagated; four studies focalized their attention on psychiatric medications and climate condition; and, finally, one study was interested in the effects of MDMA during the ambient temperature changing.

The findings extracted from the analysis of 13 papers indicate the significant impact of climate change on individuals with mental health disorders and substance abuse issues. Specifically, these papers illustrate how higher temperatures and extreme weather events pose substantial challenges, especially for vulnerable populations [25,32,33,34].

Each article studied was illustrated in Table 1 with a summary containing information about the study’s year, drugs cited, results, and conclusion. Additional information was available in Appendix A, attached as Appendix A, where the entry “sample size” is listed.

All the studies considered extreme adverse climate change, focusing on heatwaves, and its effects, both physiologically and pathologically.

## 4. Discussion

The introduction to this scoping review emphasizes that only 13 studies examining a potential correlation between substance abuse and climate change were identified. Within this context, it is noted that the limited available literature displays heterogeneity, as the studies investigate various aspects related to the correlation between pathophysiology, climate change, and substance abuse, each with distinct focuses.

Since most studies have identified correlations between higher ambient temperatures and substance use, it suggests that climate change has a substantial impact on these behaviors. However, the results often fail to consider various subgroups, despite evidence showing that climatic conditions may affect vulnerable groups differently, such as individuals with mental disorders, young people, and those with pre-existing health conditions, thereby exacerbating disparities in substance abuse exposure and outcomes during extreme weather events [4,25,43].

When examining the effects of elevated environmental temperatures, it becomes evident that such conditions can induce heat stress—a spectrum of physical ailments caused by the body’s inability to adjust to high temperatures, as outlined by Cusack et al. [17]. Symptoms may include dehydration, rash, cramps, syncope, and exhaustion.

Nonetheless, the regulation of body heat involves various mechanisms such as conduction, convection, radiation, and evaporation, which necessitate a well-maintained integumentary system and functional autonomic nervous system for efficacy [44,45].

Even though studies with psychiatric patients have explicitly been excluded from our research strategies, the overall results still pointed out repeatedly that people with mental health issues represent an especially vulnerable group in this context; several studies highlighted in this review emphasize that certain groups, such as individuals with substance addiction or mental health disorders, face heightened risks during periods of extreme heat, particularly during heatwaves [18,25]. Cusack et al. [17] elucidate the physio-pathological mechanisms underlying the impact of adverse weather on mental health. Firstly, consideration must be given to potential physical pathologies associated with mental illness [46]. Secondly, patients with mental conditions may be more vulnerable due to the effects of prescribed psychiatric medications, particularly neuroleptic and anticholinergic medications [17,47,48]. Finally, their physical or mental conditions could significantly affect their ability to cope with and adapt to adverse weather conditions [17].

Additionally, individuals must respond appropriately to environmental changes by maintaining proper hydration, abstaining from alcohol consumption, and wearing suitable clothing [17].

Cusack et al. identify vulnerable groups as being particularly susceptible to substance abuse during extreme weather conditions [17,35,39]. Factors such as age, gender, income, and existing health conditions can exacerbate these impacts and heighten the risk of substance abuse. Furthermore, climate change exacerbates mental health disorders like anxiety, depression, and PTSD, consequently increasing the likelihood of substance abuse [18,35,49].

Considering individual susceptibility, physiological changes associated with age or pathology, such as hypothalamic, cardiac, pulmonary, or renal dysfunction, could significantly impair the body’s coping mechanisms, posing a threat to life [10]. Moreover, several authors highlighted that the psychiatric and psychological implications of heat-related conditions must be considered, especially given that individuals with mental illness are typically at a higher risk [47]. This study also underscores the critical importance of addressing the intersection of climate change, mental health, and substance abuse during extreme weather events.

Even though we did not search for them, a lot of studies deal with heat effects on psychiatric populations; it is clarified that these were considered within the studies in the context of substance misuse and general substance use, rather than in studies exclusively targeting psychiatric populations.

The exclusion of studies focused on the psychiatric population, which is deferred to other comprehensive works, can be considered a relative limitation of the study.

Parks RM et al. [38], Hensel M et al. [25], and Page LA et al. [35] express serious concerns regarding the link between alcohol abuse and heatwaves in their respective studies. Page LA et al. reveal a 4.9% increase in the risk of death per 1.8 °C rise in temperature among vulnerable subjects such as younger patients and those with mental pathologies. According to Parks et al., this issue also affects cannabis consumers, increasing visits to the hospital emergency room. The data unequivocally show a troubling correlation between hotter weather and an uptick in emergency room visits related to drug overdoses [19,37,42]. As temperatures increase, there is a corresponding rise in hospital admissions due to drug-related issues such as amphetamine, cocaine, and opioid abuse [19,37,40,42].

In the United States alone, substance use disorders present a significant public health challenge, affecting millions of individuals and resulting in a considerable number of drug-related overdose deaths annually [41]. The opioid crisis has notably contributed to a significant increase in mortality rates in the USA, prompting government interventions to restrict medication prescriptions and distribution, and promote the use of buprenorphine, methadone, and naltrexone as treatment options [40,50].

However, Kilbourne M et al. suggest that the increase in overdose-related deaths is not directly correlated with cocaine abuse due to adverse weather; rather, it may be associated with the combined effect of heat stress and metabolic damage from substance abuse [42]. Nonetheless, the implicit correlation between substance abuse, particularly cocaine, and extreme weather, leading to an increase in hospitalizations and fatalities, cannot be ignored. Subsequently, Chang et al. extensively examined the effects of opioids, cocaine, and amphetamines and their correlation with the exposure to high temperatures, highlighting various side effects resulting from the impacts on thermoregulation and alertness, as well as chronic health conditions related to substance abuse. Regarding opioid abuse, Ezell JM et al. [19], consistent with the findings of Hensel M et al., describe how climate change could exacerbate opioid addiction and increase drug consumption. However, all the authors concurred on defining a correlation between opioid abuse and heatwaves in terms of susceptibility and vulnerability due to the physiological interaction between thermoregulation and drug effects. Nonetheless, Chang et al. underscore the need for more extensive and comprehensive studies to analyze the correlation between substance abuse, chronic conditions related to abuse, and adverse weather conditions.

Understanding these environmental factors is crucial for studying patterns of drug use and comprehending how external influences can modify drug-taking behaviors [32].

Adrienne et al. notably highlight the significant impact of climate change on the mental health of children and adolescents, with acute, subacute, and chronic events leading to direct effects such as trauma, indirect effects like distress resulting from forced migration, and physical alterations [39,51].

In this context, Hrabok et al. identify a robust correlation between substance and alcohol abuse following natural disasters, especially among individuals experiencing mental health conditions such as PTSD or severe anxiety developed post-disaster, particularly in young people [18,52,53].

These findings underscore the interconnectedness of climate change and drug use and the necessity of studying and intervening to address this significant issue. In summary, the papers reviewed encompassed observational research, ecological studies, qualitative discussions, and case series investigations, integrating the realms of environmental change and public health outcomes related to substance abuse.

Climate change can exacerbate existing health problems, particularly for individuals with mental health disorders, substance abuse, or specific medical needs [35,39]. Very high temperatures and intense humidity place a significant strain on the body, increasing the susceptibility to illness or even death.

The detailed analyses and conclusions drawn from these studies highlight the urgent need for diverse healthcare interventions, proactive strategies, and collaborative efforts across disciplines to address the complex interplay between climate change, heat exposure, and substance abuse. It necessitates collaboration among healthcare professionals, policymakers, and communities to build resilience and protect the most vulnerable individuals from climate-related challenges. In essence, understanding and addressing the links between climate change, mental health, and substance abuse are crucial for safeguarding people’s well-being in a changing climate.

Furthermore, the research has identified a significant number of initial works. The high volume of literature is likely due to the inclusion of general terms that were necessary for comprehensive coverage. It should be noted as a study limitation that specific terms related to climate-change-induced phenomena were deliberately omitted to minimize irrelevant literature.

This scoping review identifies only 13 studies exploring the link between substance abuse and climate change. The literature is limited and heterogeneous, with studies examining various aspects of this correlation. Most studies indicate correlations between higher ambient temperatures and substance use, highlighting significant impacts of climate change on these behaviors. However, the research often overlooks diverse subgroups, such as individuals with mental disorders, young people, and those with pre-existing health conditions, potentially exacerbating disparities in substance abuse outcomes during extreme weather events. Specifically, the review predominantly focuses on studies investigating heatwaves and high ambient temperatures, rather than other extreme weather events like floods and other phenomena.

### 4.1. Sociological Aspects

It is important to note that this paragraph does not refer to the findings of this study but aims to clarify that the underlying climate change entails a multiplicity of complex factors briefly mentioned here. However, these factors lie beyond the scope of our biomedical and toxicological research focus.

Climate change not only threatens global ecosystems but also deeply influences socio-economic dynamics, including the consumption and trade of drugs. According to the IPCC report of 2021, climate change is responsible for significant variations in weather patterns and the distribution of natural resources, resulting in impacts on agriculture, fisheries, and food security worldwide [5]. These environmental disruptions can have direct effects on drug cultivation, altering the availability and quality of raw materials necessary for the production of narcotics [54].

Climate instability increases the vulnerability of rural communities dependent on illicit cultivation, making them more susceptible to food and economic crises that can promote the adoption of alternative survival strategies, including the intensification of drug trafficking [55,56]. These changes can alter the patterns of drug consumption and distribution, influencing both local communities and the global narcotics market [28].

Addressing the intricate interplay among climate change, illicit drug economies, and socio-cultural dynamics necessitates an integrated, multidisciplinary approach. Engaging experts from policy, environmental sciences, economics, and public health is crucial for developing effective strategies that mitigate the adverse impacts of climate change on drug consumption and trafficking, while fostering resilience among vulnerable communities [5,26,56].

### 4.2. Limits

This scoping review aimed to comprehensively explore the intersection of substance abuse and climate change; several limitations warrant acknowledgment:-The exclusion of psychiatric disorders: The exclusion of studies focused on psychiatric populations, deferred to other comprehensive works, may be considered a relative limitation of this study;-Limitations in the search methodology: Despite employing detailed and tailored search strategies, there could have been potential bias in the article selection, especially given the specificity of the search terms and the restriction to English-language publications;-Limited geographical scope: The predominance of studies conducted in the United States may restrict the generalizability of findings to different geographic contexts, considering the regional and cultural variations in climate.-The exclusion of other disciplines: The exclusion of articles not specifically addressing substance abuse in relation to climate change may have limited the breadth of perspectives explored in this study.-The quality assessment of studies: Despite a systematic approach to review, the subjective nature of assessing the quality of the included studies could impact the overall reliability of the conclusions.

## 5. Conclusions

The scoping review presented a comprehensive analysis of 13 studies examining the correlation between substance abuse and climate change, revealing a significant impact on vulnerable populations, particularly those with mental health disorders. The heterogeneity of the literature underscores the complexity of this correlation, with studies exploring various aspects related to pathophysiology, climate change, and substance abuse.

The findings highlight the substantial challenges posed by higher temperatures and extreme weather events, exacerbating substance abuse issues, particularly during heatwaves. Vulnerable groups, including individuals with mental health conditions, are at a heightened risk, necessitating a holistic approach to address their unique needs.

This review demonstrates that the correlation between substance abuse and climate change at various levels is still relatively unexplored, although the existing literature may provide valuable insights for further exploration by the scientific community.

To better understand and address the impact of climate change on substance abuse, we recommend the following actions: expanding the research to explore how environmental changes affect drug production, distribution, and consumption; fostering multidisciplinary approaches involving climatologists, sociologists, public health experts, and economists; conducting longitudinal studies to monitor substance abuse trends in relation to climate variability; assessing the effectiveness of current policies and proposing evidence-based interventions to protect vulnerable populations; and implementing community-specific programs to enhance resilience to climate change and reduce substance abuse, utilizing innovative technologies such as AI to effectively respond to emerging challenges.

## Figures and Tables

**Figure 1 ijerph-21-00896-f001:**
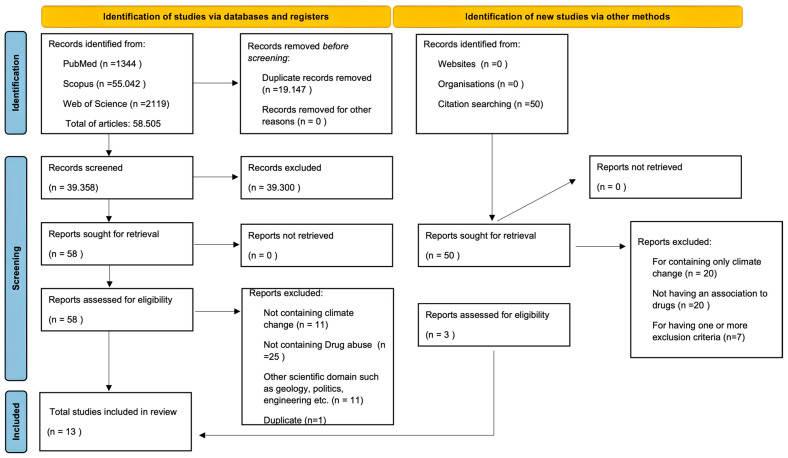
Diagram obtained through Prisma 2020 regarding the articles identified and screened.

**Table 1 ijerph-21-00896-t001:** Summary table regarding the selected works within the scope of the scoping review.

Study	Place	Interval	Drugs Analyzed	Climate Change Event	Type of Study	Observations
Cusack et al. (2011) [17]	Australia	1985–2010	Opioids, alcohol, diazepam, and amphetamines	Heatwaves	Discussion paper	Medications associated with heat intolerance and positive association of high risk of heat-related illnesses during heatwaves.
Hrabok et al. (2020) [18]	Canada, Alberta	2020	Alcohol and substance abuse	Wildfire	Qualitative	N/A
Ezell et al. (2023) [19]	USA, California	2010–2020	Opioids, methamphetamines, crack, cocaine, GHB/GB, ketamine, PCP, heroin, ecstasy, and methamphetamine	Climate change effects on PWUDs	Qualitative	Extreme heat enhances sedative effects of opioids and increases body temperature with stimulant use, raising overdose and adverse effect risks.
Hensel et al. (2021) [25]	Germany, Hamburg	2010–2014	Alcohol, opioid, sedatives/hypnotics, and psychoactive substances	Ambient temperature	Prospective, observational	Hot weather associated with increased frequency of severe acute poisoning by alcohol and drugs.
Aarde et al. (2017) [32]	N/A	N/A	MDMA	High and low ambient temperatures	Experimental	Increased acquisition of MDMA self-administration in test subjects at higher ambient temperatures.
Page et al. (2012) [35]	UK	1998–2007	Antipsychotics, antidepressants, and hypnotics/anxiolytics	Heatwaves, increased temperatures	Observational, quantitative	Hot weather increases relative risk of death, especially in younger patients and those with substance or alcohol misuse.
Chang HH et al. (2023) [36]	USA, California	2005–2019	Amphetamine, cocaine, and opioids	Daily ambient temperature	Observational, quantitative	Positive association between higher temperatures and emergency department visits for stimulant and opioid use, and other drugs.
Marzuk et al. (1998) [37]	USA, NY	1990–1995	Cocaine, benzoylecgonine, opiates, ethanol, and various other drugs,	Hot weather	Retrospective review, quantitative	High ambient temperature increases mortality from cocaine overdose, especially above 31.1 °C.
Parks et al. (2023) [38]	USA, NY	1995–2014	Cannabis, cocaine, opioids, and sedatives	Rising temperatures	Observational, quantitative	Higher temperature linked to increased hospital visits for alcohol- and substance-related disorders.
van Nieuwenhuizen A et al. (2021) [39]	USA, California	June 2021	Antipsychotics	Extreme weather, temperature increase	Review, quantitative	Children and adolescents at increased risk of hyperthermia during extreme heat events.
Ryus et al. (2022) [40]	USA, Connecticut	2019	Opioids and benzodiazepines	Heatwave	Qualitative, case series	Extreme heat exacerbates health risks, leading to dehydration, lethargy, and increased mortality in opioid users.
Reser et al. (2020) [41]	USA	2014–2019	Not drug-specific	N/A	Quantitative	N/A
Kilbourne et.al. (1998) [42]	USA, NY	1993–1995	Cocaine	Heatwaves	Quantitative	Significant rise in deaths from cocaine overdose during hot weather, i.e., impact of extreme temperatures on drug-related fatalities in urban settings like New York City during heatwaves.

## Data Availability

The data utilized for this Scoping Review are drawn from a comprehensive search encompassing various databases and sources. All search terms employed in this review are detailed in the methodology section. The included studies, references, and relevant information extracted are publicly accessible and retrievable through the respective databases or repositories referenced in the bibliography.

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
