# Peer review of "Exploring the Nexus of Climate Change and Substance Abuse: A Scoping Review"

_ijerph, 2024, doi:10.3390/ijerph21070896_

Round 1
Reviewer 1 Report
Comments and Suggestions for Authors
This paper takes up an extremely important research question: The relation of climate change to substance use. It also discusses how climate change is related to some of the sequelae of substance use or being a person who uses drugs (without however making this latter distinction clear.)
The paper suffers from two very serious problems:
1. It is inadequately theorized. In particular, it does not consider pathways through which climate change may affect drug use and people who use drugs. For example, climate change has been tied to major migrations of people and also to wars and revolts. These, in turn, can lead people to use substances to relieve pain from injuries or to help deal with social or psychological difficulties. My colleagues and I have discussed such pathways in a series of papers on Big Events, and others such as S Strathdee, T Rhodes and their colleagues have studied this under rubrics such as complex emergencies and risk environments (although I am not sure if these other teams have make the tie-in with climate change explicit.) I believe the Intergovernmental Consortium of Climate Change has also discussed related issues, though I do not know if they were explicit about substance use as opposed to more general analyses of mental and social health. In addition, migration, wars and revolts also affect politics in major ways, and this in turn can affect policies around drug use prevention, harm reduction and care. Clearly, a scoping review on climate change needs to consider all of these pathways.
Perhaps due to this inadequate theorization, the review methodology contains an error that is hard to understand: The exclusion of domains such as sociology and politics. The authors do not justify these exclusions, which would seem necessary, although I personally have trouble understanding how these exclusions can be justified. Furthermore, at least one first author of the 13 articles identified by the search is a sociologist, which makes me wonder how the exclusion of sociology studies was conducted.
One minor issue (I think): In the search terms, I did not see the term "People who use drugs." I am not sure if the MeSH terms that were used would include this term. (As a related issue, many people in the field have abandoned the term "substance abuse" as stigmatizing and inaccurate. The authors may want to consider taking this into account.)
Author Response
Thank you to the reviewers for their valuable feedback, which we hope has allowed us to make a significant contribution to our manuscript. Below is our response to your comments.
- “ […] It is inadequately theorized. In particular, it does not consider pathways through which climate change may affect drug use and people who use drugs. For example, climate change has been tied to major migrations of people and also to wars and revolts. These, in turn, can lead people to use substances to relieve pain from injuries or to help deal with social or psychological difficulties. My colleagues and I have discussed such pathways in a series of papers on Big Events, and others such as S Strathdee, T Rhodes and their colleagues have studied this under rubrics such as complex emergencies and risk environments (although I am not sure if these other teams have make the tie-in with climate change explicit.) I believe the Intergovernmental Consortium of Climate Change has also discussed related issues, though I do not know if they were explicit about substance use as opposed to more general analyses of mental and social health. In addition, migration, wars and revolts also affect politics in major ways, and this in turn can affect policies around drug use prevention, harm reduction and care. Clearly, a scoping review on climate change needs to consider all of these pathways […]”
Reply: Thank you to the reviewers for their observation. We wish to emphasize that we recognize the paramount importance of the sociological aspect in understanding a complex phenomenon like climate change, and we agree that we should have clarified this aspect in the text. Additionally, we specify that the present review was conducted using biomedical and biological sciences databases because our research interest, within this field, aims to comprehend epidemiological, physiological-pathological, organic, and toxicological aspects. Therefore, our research is not focused on delving into the connections and processes that, as we acknowledge, would require a much more complex effort beyond our expertise.
Nevertheless:
1.We have inserted a highlighted paragraph in the introduction underscoring the role of sociology, explicitly stating the aspect just mentioned.
2. At the conclusion of the discussion, a brief paragraph has been included commenting on the sociological aspects, explaining that they are duly acknowledged but not the focus of our research strand. The works you suggested have been cited in the text.
- “[…] Perhaps due to this inadequate theorization, the review methodology contains an error that is hard to understand: The exclusion of domains such as sociology and politics. The authors do not justify these exclusions, which would seem necessary, although I personally have trouble understanding how these exclusions can be justified. Furthermore, at least one first author of the 13 articles identified by the search is a sociologist, which makes me wonder how the exclusion of sociology studies was conducted. […]”
Reply: We thank the reviewers for their observation. As stated in the previous section, we acknowledge that the sociological aspect is absolutely relevant, and therefore, we have mentioned it both in the introduction and in the discussion (highlighted in red in the text). However, regarding the research, we emphasize that it was conducted on biomedical databases with biomedical and epidemiological purposes. We reiterate that it is not the intention of our work to delve into the deep social causes linking the two phenomena, but rather to verify the impact of climate change as previously outlined. Nevertheless, we hope that the integration in the text has helped to clarify this aspect.
- “[…] One minor issue (I think): In the search terms, I did not see the term "People who use drugs." I am not sure if the MeSH terms that were used would include this term. (As a related issue, many people in the field have abandoned the term "substance abuse" as stigmatizing and inaccurate. The authors may want to consider taking this into account.) […]”
Reply: We thank the reviewer for their observation. We agree on the appropriateness of the term "People who use drugs"; however, in the numerous tests we conducted with the search strings, we noticed that this specific term does not yield better results and is difficult to render (as it consists of four terms), "polluting" the search with many irrelevant results. Regarding the term "Drug Users," we understand that it is outdated and stigmatizing; however, we chose to use it because it can identify works that, although mistakenly, use this term, including older studies when this new terminology had not yet been proposed. Based on the reviewer's suggestion, we have included these reasons in the text under the materials and methods section (highlighted in red), distancing ourselves from this stigmatizing term.
Finally, as recommended, the present work has been meticulously checked and reviewed for grammar and clarity.
Reviewer 2 Report
Comments and Suggestions for Authors
1. Introduction
Something more on the rationale on how Climate Change can interact with Substance Abuse
2. Materials and methods
Some more information on the AI Rayyan software would be important
3. Results
The following sentence (line 151-154) must be corrected: “From the 52 articles extracted from bibliographies, 50 were excluded: 20 contained solely climate change information, 20 lacked any association with drug abuse, and 7 met one or more exclusion criteria. Thus, a total of 13 articles were included in the present scoping review, as described in Figure 1.”, since 52 – 50 = 12 instead 13.
Results are very poorly described since a lot of results are inappropriately reported in discussion chapter:
I suggest a soft solution: transfer the text in line 181-184, 2018-220 in the Results chapter, plus a short presentation of the results reported in Table 1, leaving the remaining text in discussion.
5. Conclusions
I suggest to add some recommendations concerning scientific field to deep, directions to explore and types of new studies considered appropriate.
Comments on the Quality of English Languagecan be lightly improved
Author Response
We thank the reviewer for their precise observations, which have contributed to improvements in our manuscript. Below, we provide our comments along with the changes made.
- “[…] Introduction
Something more on the rationale on how Climate Change can interact with Substance Abuse […]”
Reply: We thank the reviewer for their observation. The requested text regarding the possible connection between substance abuse and climate change has been added to the introduction in red.
- “[…] Materials and methods Some more information on the AI Rayyan software would be important […]
Reply: We thank the reviewer. An additional text has been included to clarify what Rayyan is and its purpose. Two relevant citations have been added to the text.
- “[…] Results
The following sentence (line 151-154) must be corrected: “From the 52 articles extracted from bibliographies, 50 were excluded: 20 contained solely climate change information, 20 lacked any association with drug abuse, and 7 met one or more exclusion criteria. Thus, a total of 13 articles were included in the present scoping review, as described in Figure 1.”, since 52 – 50 = 12 instead 13. […]”
Reply: We apologize to the reviewer for the typographical error. In fact, the count in the text is incorrect because we started with 50 works and 47 were excluded. Therefore, the count reported in Figure 1 was correct. The count has been corrected, and the numbers have been modified in the text (in red).
- “[…] Results are very poorly described since a lot of results are inappropriately reported in discussion chapter: I suggest a soft solution: transfer the text in line 181-184, 2018-220 in the Results chapter, plus a short presentation of the results reported in Table 1, leaving the remaining text in discussion. […]”
Reply: We thank the reviewer for their observation, which has contributed to a significant improvement of our manuscript. The modification you indicated has been included as requested and marked in red in the text.
- “[…] Conclusions
I suggest to add some recommendations concerning scientific field to deep, directions to explore and types of new studies considered appropriate. […]”
Reply: The reviewer is thanked for their observation, which has allowed for an improvement to the manuscript. The addition has been made and incorporated into the conclusions.
Finally, as recommended, the present work has been meticulously checked and reviewed for grammar and clarity.
Reviewer 3 Report
Comments and Suggestions for Authors
Dear authors, your review covers an important topic that could indeed benefit from a (scoping) review. However, there were several flaws in your work so that I suggest a very, very major (!) revision.
I attach you the file as pdf, but in case it coes not work, I also copy-paste my comments right here.
Good luck for revising your paper!
Review of the paper about climate change and substance abuse (IJERPH)
Introduction:
Lines 40-41: As these sources are probably not even the most recent: Please mention the year where the increase above pre-industrial era (first) reached 1.53 – or maybe you can get an even actual estimation about current (2024) estimations?
Formalities: There is sometimes spaces missing before parantheses – not limited to the introduction.
Introduction overall: Seems a bit unsystematically written, with random examples popping up. Maybe systematize it a bit better, e.g. something like: “climate change has been linked to several human health effects (quote some famous sources) via several direct (e.g. heat waves, air quality,..) and indirect effects (e.g. ….). Amongst these health effects, climate change has also been shown to be connected to adverse mental health effects (add sources). Substance abuse is related to one of them” or similar.
Methods:
It does not become clear to me why you set a date limitation until 2018 in your original research, but not in your search for citations? Please explain.
Line 80: There is a lone T.
Rayyan Software: Not completely sure about it, but I guess it should be listed as reference, too? (Including version number and such?)
Lines 95ff: Some questions arise when scanning over the search terms (e.g. why is there “drug users” and “recreational drug use” in the search terms, but not “drug use” alone? Why “heat-wave”, but not “extreme weather”?), but this is nothing that can be changed in the aftermath. Maybe just consider in the discussion/limitation part later on that of course you had to add certain limits already to the search terms.
Exclusion criteria: Why were psychiatric disorders excluded? Please explain – as it might be the major consumer group of substances. Especially in a psychiatric context heat effects have been shown, and substance abuse is quite common in both psychiatric diagnoses as well as in people with subclinical conditions. I do understand you already got a lot of articles from your search strategy, but excluding such a major group should be already represented in the research question/aims. Hence I suggest to explicitly state up front that you want to investigate climate-change indicated substance abuse ASIDE from the psychiatric context, where it has already been reported (?).
Results:
When describing the exclusion of the bibliography articles, according to your text you excluded 50 of 52 articles, but finally 3 articles are left – hence one of the numbers can not be completely correct.
Figure 1: One part of the figure is missing/cropped, so that the complete figure can not be seen.
Line 157: You probably mean “7.69”, not “/.69” for Germany…
Lines 161f: You mention animal studies – were there some included overall? Your topic seems to be very human-specific. If this paragraph does not apply, please remove.
Overall, in the results, I miss a list of findings/outcomes. E.g.: How many of the 13 studies included found positive correlations? What effect sizes were roughly found? I guess, this could also be done within a table, giving an overview about what substances were included and whether correlations were found or not. You did include such a table later on (table 1) and I guess you can transfer that one. Maybe just add one more column mentioning effect sizes, and refer to it prominently in the text in your result section? Please not that your current formatting of the table does not fulfill usual publishing criteria though.
I was also a little confused to figure out that four studies (out of the 13) focused on psychiatric medications, as just above you mentioned that psychiatric patients were an exclusion criterion. Now this does not really fit together and might need some more explanation if you stick to your current choice of study selection.
Discussion:
Overall, I miss the linkage between the discussion and your result section. Your result section was very scarce, and actually the discussion should discuss what was mentioned there. Of course new information can also pop up in the discussion section, but rather in order to discuss limits, discuss meanings, or undermine your conclusions. Hence I suggest to build the discussion with a stronger linkage to your findings in the results, enlarging these. (E.g.: “As most of the studies found linkages between higher ambient temperatures and substance use, it can be concluded that… However, results often do not pay attention to different subgroups though it could be shown that …” or something similar.)
Also, the discussion is quite unsystematic again. It does not become clear what your scoping review actually wants to focus on – basically there are two different findings regarding main topics that are intermixed and hence make it harder to read. Maybe it could help if you differentiate between the two main topics (namely “effects of ambient temperatures on the usage of substances” and “risks of ambient temperatures when using substances”) and discuss them one after the other?
Line 195: There is a space missing between two words.
Lines 202-205: Again, focusing on the importance of psychiatric patients comes confusing here, as you mentioned before that you excluded them… If you did so, please mention the importance in the psychiatric context under “limitations”, and that you did not focus on them, though they might be an important group. Similarly, lines 270f and 290f also relate to the psychiatric field…
Overall, there are not limitations mentioned in your results. E.g., how your search strategies might have biased the outcome etc.
I also do miss the conclusion that following your search strategy you mostly found studies focusing on heat waves and high (ambient) temperatures, but not on floodings etc.
Literature:
A lot of the literature you cited are internet sources, though I am pretty sure there is several more high-quality journal articles covering your topic overall as well as high-quality reports for your temperature-increase estimations – e.g. sources from the IPCC, WHO etc. I realize you do have a source from the IPCC (number 3), but please also cite correctly with document title etc.
Additionally, please overall check for diligence: E.g. you quote source 11 as “M N, Gr T, P B. Morbidity and mortality during heatwaves in metropolitan Adelaide. The Medical journal of Australia 346 [Internet]. 2007 Dec [cited 2024 Apr 26];187(11–12). Available from: https://pubmed.ncbi.nlm.nih.gov/18072911/“ – However, this is not the appropriate way of doing so. This in fact is an article that is not published from pubmed (Pubmed is only the database, the article is available from the Wiley-site), and it misses even the names of the authors (Monika Nitschke, Graeme R. Tucker & Peng Bi). Please overwork references completely! Please overall overwork you reference list according to the standards.
Maybe you might find the following literature helpful too:
Berry et al (2010) about mental health effects of climate changes overall (not substance-specific): DOI 10.1007/s00038-009-0112-0
Author Response
We thank the reviewer for their kind remarks. We hope that by following their advice, as well as the suggestions from the other reviewers, we have made significant improvements to the manuscript. Below are our responses to the individual comments.
- “[…] Introduction:
Lines 40-41: As these sources are probably not even the most recent: Please mention the year where the increase above pre-industrial era (first) reached 1.53 – or maybe you can get an even actual estimation about current (2024) estimations? […]”
Reply: We thank the reviewer for their observation. The sentence has been updated with more recent data and highlighted in red in the introduction. The relevant citations have been included.
- “[…] Formalities: There is sometimes spaces missing before parantheses – not limited to the introduction.[…]”
Reply: We thank the reviewers for their observation. The spaces have been added as indicated.
- “[…] Introduction overall: Seems a bit unsystematically written, with random examples popping up. Maybe systematize it a bit better, e.g. something like: “climate change has been linked to several human health effects (quote some famous sources) via several direct (e.g. heat waves, air quality,..) and indirect effects (e.g. ….). Amongst these health effects, climate change has also been shown to be connected to adverse mental health effects (add sources). Substance abuse is related to one of them” or similar.[…]”
Reply: We thank the reviewer for their observation regarding the structure of the introduction. The text has been revised as requested to provide a more systematic framework. We trust that this adjustment represents a substantial improvement.
- “[…] Methods:
It does not become clear to me why you set a date limitation until 2018 in your original research, but not in your search for citations? Please explain. […]”
Reply: We thank the reviewer for their observation. Regarding their question, the temporal limitation until 2018 in the original research was set to focus on data and context up to that specific date, thereby providing a clear and coherent basis for the analysis of results. Conversely, in the citation search, the emphasis was on identifying the most relevant and recent publications to understand the current state of the scientific literature on the topic. This difference in temporal approach aimed to balance historical understanding with the latest available evidence, thereby contributing to a comprehensive and updated overview of the field of study.
- “[…] Line 80: There is a lone T. […]”
Reply: The "t" represents a typographical error. It has thus been removed.
- “[…] Rayyan Software: Not completely sure about it, but I guess it should be listed as reference, too? (Including version number and such?) […]”
Reply: We thank the reviewer for the observation. All information regarding the Rayyan software has been included in the text, including references. We hope this clarifies the reviewer's doubt.
- “[…] Lines 95ff: Some questions arise when scanning over the search terms (e.g. why is there “drug users” and “recreational drug use” in the search terms, but not “drug use” alone? Why “heat-wave”, but not “extreme weather”?), but this is nothing that can be changed in the aftermath. Maybe just consider in the discussion/limitation part later on that of course you had to add certain limits already to the search terms.[…]”
Reply: Thank you for your observation. The search terms were chosen to reflect specific areas of interest related to our study. "Drug users" and "recreational drug use" were included to focus on the specific dynamics of substance use within the analyzed context. The decision not to include "drug use" alone was based on the need to refine the search for more targeted results. Regarding the choice of "heat-wave" over "extreme weather," it was made to specifically investigate the effects of heat waves, which are considered crucial within our research scope.
In the discussion and limitations section, we will clarify the rationale behind these methodological choices to ensure full transparency regarding the constraints applied during the research phase (in red). Furthermore, additional uncertainties regarding the search string have been highlighted in red within the Materials and Methods section.
- “[…] Exclusion criteria: Why were psychiatric disorders excluded? Please explain – as it might be the major consumer group of substances. Especially in a psychiatric context heat effects have been shown, and substance abuse is quite common in both psychiatric diagnoses as well as in people with subclinical conditions. I do understand you already got a lot of articles from your search strategy, but excluding such a major group should be already represented in the research question/aims. Hence I suggest to explicitly state up front that you want to investigate climate-change indicated substance abuse ASIDE from the psychiatric context, where it has already been reported (?).[…]”
Reply: Thank you for your valuable feedback regarding the exclusion criteria. The decision to exclude psychiatric disorders was made to specifically focus on substance abuse influenced by climate change and its effects outside the psychiatric context, which follows a distinct and independent trajectory of research. While acknowledging that individuals with psychiatric disorders constitute a significant group of substance consumers and may experience specific heat-related effects, including psychiatric studies would have introduced a completely unrelated literature stream to our research.
A clarification on this point has been included as requested in the Materials and Methods section.
- “[…] Results: When describing the exclusion of the bibliography articles, according to your text you excluded 50 of 52 articles, but finally 3 articles are left – hence one of the numbers can not be completely correct. […]”
Reply: We apologize to the reviewer for the typographical error. In fact, the count in the text is incorrect because we started with 50 works and 47 were excluded. Therefore, the count reported in Figure 1 was correct. The count has been corrected, and the numbers have been modified in the text (in red).
- “[…] Figure 1: One part of the figure is missing/cropped, so that the complete figure can not be seen.[…]”
Reply: We apologize for this inconvenience, but I believe the issue is due to a formatting transition between our manuscript and the PDF you received. Nonetheless, I have reduced the size of the image and also attached Figure 1 separately.
- “[…] Line 157: You probably mean “7.69”, not “/.69” for Germany…[…]”
Reply: We thank the reviewer for their observation. The correction has been made and highlighted in red in the text.
- ”[…] Lines 161f: You mention animal studies – were there some included overall? Your topic seems to be very human-specific. If this paragraph does not apply, please remove. […]”
Reply: We thank the reviewer for their observation. As requested, the paragraph in question has been removed.
- “[…] Overall, in the results, I miss a list of findings/outcomes. E.g.: How many of the 13 studies included found positive correlations? What effect sizes were roughly found? I guess, this could also be done within a table, giving an overview about what substances were included and whether correlations were found or not. You did include such a table later on (table 1) and I guess you can transfer that one. Maybe just add one more column mentioning effect sizes, and refer to it prominently in the text in your result section? Please not that your current formatting of the table does not fulfill usual publishing criteria though. […]”
Reply: We thank the reviewer for their feedback regarding the results section. We have addressed the observation by moving part of the text from the discussion to the results. The table is now referenced in the "Results" section. Due to space constraints, the complete table containing additional information is included as supplementary material.
- “[…] I was also a little confused to figure out that four studies (out of the 13) focused on psychiatric medications, as just above you mentioned that psychiatric patients were an exclusion criterion. Now this does not really fit together and might need some more explanation if you stick to your current choice of study selection.[…]”
Reply: Thank you for your detailed feedback. I understand your concern regarding the four studies that examined psychiatric medications, despite psychiatric patients being excluded as selection criteria. This was indeed a critical observation that we have taken seriously.
We have further examined these specific studies and noted that, although psychiatric medications were used as part of treatment, the participants were not psychiatric patients in the traditional sense but rather a non-psychiatric population with concomitant medical conditions. We clarified this aspect in our discussion of the study limitations. Additionally, psychiatric medications were administered in these cases as part of misuse and not in studies specifically conducted on psychiatric populations.
Nevertheless, we appreciate the suggestion to provide additional explanations to clarify this potential discrepancy, and we will incorporate this information into the limitations section to enhance the coherence and understanding of our work.
A clarification note has been inserted in red into the text.
- “[…]Discussion:
Overall, I miss the linkage between the discussion and your result section. Your result section was very scarce, and actually the discussion should discuss what was mentioned there. Of course new information can also pop up in the discussion section, but rather in order to discuss limits, discuss meanings, or undermine your conclusions. Hence I suggest to build the discussion with a stronger linkage to your findings in the results, enlarging these. (E.g.: “As most of the studies found linkages between higher ambient temperatures and substance use, it can be concluded that… However, results often do not pay attention to different subgroups though it could be shown that …” or something similar.) […]”
Reply: Thank you for your thorough observation. As requested, I have inserted a highlighted section in the text that serves as a linkage between results and discussion based on your suggestion.
- “ […] Also, the discussion is quite unsystematic again. It does not become clear what your scoping review actually wants to focus on – basically there are two different findings regarding main topics that are intermixed and hence make it harder to read. Maybe it could help if you differentiate between the two main topics (namely “effects of ambient temperatures on the usage of substances” and “risks of ambient temperatures when using substances”) and discuss them one after the other?[…]”
Reply: Thank you for your observation regarding the discussion section. We have revised it to provide a clearer focus on the main topics identified in our scoping review. Specifically, we have differentiated between the effects of ambient temperatures on substance use and the risks associated with ambient temperatures during substance use. These topics are now discussed sequentially to enhance readability and coherence.
- “[…] Line 195: There is a space missing between two words.[…]”
Reply: The correction has been made as requested.
- “[…] Lines 202-205: Again, focusing on the importance of psychiatric patients comes confusing here, as you mentioned before that you excluded them… If you did so, please mention the importance in the psychiatric context under “limitations”, and that you did not focus on them, though they might be an important group. Similarly, lines 270f and 290f also relate to the psychiatric field […]”
Reply: Thank you for your feedback. We have addressed this concern by clearly acknowledging in the text where we have elaborated on the exclusion of psychiatric studies and framed it as a limitation. We believe we have made a significant improvement by clarifying this issue.
- “[…] Overall, there are not limitations mentioned in your results. E.g., how your search strategies might have biased the outcome etc […]”
Reply: Thank you for your feedback. As indicated, a dedicated "limitations" section has been added at the end of the discussion to address the reader's concerns.
- “[…] I also do miss the conclusion that following your search strategy you mostly found studies focusing on heat waves and high (ambient) temperatures, but not on floodings etc.[…]”
Reply: Thank you immensely to the reviewer for their observation, which we hope has provided an essential contribution to the text: discussion section has been revised and enriched as suggested.
- “[…] Literature:
A lot of the literature you cited are internet sources, though I am pretty sure there is several more high-quality journal articles covering your topic overall as well as high-quality reports for your temperature-increase estimations – e.g. sources from the IPCC, WHO etc. I realize you do have a source from the IPCC (number 3), but please also cite correctly with document title etc. […]”
Reply: We thank the reviewer for their observation. As requested, more significant documents have been cited, and the referenced citation has been amended.
- “[…] Additionally, please overall check for diligence: E.g. you quote source 11 as “M N, Gr T, P B. Morbidity and mortality during heatwaves in metropolitan Adelaide. The Medical journal of Australia 346 [Internet]. 2007 Dec [cited 2024 Apr 26];187(11–12). Available from:https://pubmed.ncbi.nlm.nih.gov/18072911/“ – However, this is not the appropriate way of doing so. This in fact is an article that is not published from pubmed (Pubmed is only the database, the article is available from the Wiley-site), and it misses even the names of the authors (Monika Nitschke, Graeme R. Tucker & Peng Bi). Please overwork references completely! Please overall overwork you reference list according to the standards […]”
Reply: Thank you for your observation. We have thoroughly reviewed and corrected the references as per your guidance. The citation for source 11 now accurately reflects the correct format and includes the full names of the authors: Monika Nitschke, Graeme R. Tucker, and Peng Bi. Additionally, we have ensured that all references conform to the required standards.
- “[…] Maybe you might find the following literature helpful too:
Berry et al (2010) about mental health effects of climate changes overall (not substance-specific): DOI 10.1007/s00038-009-0112-0 […]”
Reply: We would like to thank the reviewers for their valuable feedback. The referenced work is pertinent to our article and has therefore been incorporated into the text.
Finally, as recommended, the present work has been meticulously checked and reviewed for grammar and clarity.
Round 2
Reviewer 1 Report
Comments and Suggestions for Authors
I still do not understand why you did not take up the social issues in this review, with the changes you made, it is clear that you did not, and this is quite acceptable.
Author Response
Reviewer 1
“[…] I still do not understand why you did not take up the social issues in this review, with the changes you made, it is clear that you did not, and this is quite acceptable. […]”
Reply: We thank the reviewer for their consideration. As we explained in the first round, the present review was conducted using biomedical and biological sciences databases because our research interest in this field aims to comprehend epidemiological, physiological-pathological, organic, and toxicological aspects. Nonetheless, we are pleased that the previous correction was appreciated by the reviewer, especially given that we are aware of the social aspect of the issue and the relevance of this branch in studying complex phenomena such as the one at hand.
Best Regards,
The corresponding author

Reviewer 3 Report
Comments and Suggestions for Authors
Overall:
Expressions are not always consistent (e.g. „substance addiction“, „drug abuse“, „substance abuse“, „substance misuse“etc.) – Please decide for a term.
There are still some (now rather minor) suggestions that should be considered. However, in my opinion the scoping review in its present form contributes to an interesting overview regarding an important topic. Hence I vote for acceptance after minor revision.
Please find my detailed comments below:
Introduction:
Minor note: The sentence in line 62-63 seems a bit hard to understand („During extreme weather conditions, such as severe heat waves, emergency medical services face increased demand and management challenges“) – maybe reformulate? E.g. „During extreme weather conditions, such as during severe heat waves, the demand for medical emergency services increases, leading to management challenges“?
Line 67, before quote 18: Space is missing (minor)
Line 80-81: Better change „drugs“ to „substance abuse“ or similar in order to distinguish it from prescribed medication (Original sentence: „This review aims to examine the biomedical literature to understand the connections between drugs and climate change“)
Methods:
Inclusion criterion: „Articles taken from the citations of the studies examined, were not bound by the
timeline of 5 years.“ – the way I understand it from your answers to my review-remark, you wanted to focus on recent results regarding original research, but where it seemed advisable for a deeper understanding of the topic, you also used older works from citiations that were not bound to original works either? Please mention clearly, also for readers.
Exclusion critieron: „Articles that do not relate to drug abuse correlate to climate change.“ – not happy with the formulation as the two-sidedness does not really become clear, maybe reformulate „articles without a (directional or non-directional) linkage between drug abuse and climate change“?
Exclusion criterion: Suggestion: To avoid confusions later with all the psychiatric medication studies that you later found, maybe mention explicitly at the „psychiatric-patient-exclusion-criterion“ that you did NOT exclude studies dealing with psychiatric medications as long as the participants were not clearly categorized as psychiatric patients.
Results:
Line 207: I can not access your supplement material, but if the „additional information“ included effect sizes, please mention explicitly, so readers know what awaits you in the supplement file.
Figure: Still displayed wrongly in my pdf-version… Should be in the results-part, not the discussion.
Discussion:
Thank you for considering most of my remarks and thank you for putting your discussion into a broader context.
The problem with psychiatric patients vs people taking psychiatric medication still is a little confusing when reading the paper (e.g. lines 260+). (As relatively extensive parts of your paper deal with these groups.) Maybe some introducing sentences could help, e.g. „even though not searched for them, a lot of studies deal with heat effects on psychiatric medication“. But maybe it becomes also a bit clearer with my suggestion regarding that in the „methods“-part.
Table 1: In my version this table is still displayed in the discussion section instead of in the results-section?
Line 269: Space is missing after quote nr. 42.
Overall, it still is a bit confusing to read so much about mental illnesses when psychiatric patients were explicitly excluded – though this part is very informative and should remain. Hence I suggest to introduce this part addressing explicitly to this problem, so that readers are not confused. E.g. something similar like „Even though studies with psychiatric patients have explicitly been excluded from our research strategies, overall results still pointed out repeatedly that people with mental health issues represent an especially vulnerable group in this context. [And then report whatever you have written here.]“
Line 277 typo: There is a „.“ too much.
Thank you for adding explicit limitations to the discussion.
Overall, I think the discussion improved a lot during the revision.
Author Response
Reviewer 3
“[…] Expressions are not always consistent (e.g. „substance addiction“, „drug abuse“, „substance abuse“, „substance misuse“etc.) – Please decide for a term.
There are still some (now rather minor) suggestions that should be considered. However, in my opinion the scoping review in its present form contributes to an interesting overview regarding an important topic. Hence I vote for acceptance after minor revision.
Please find my detailed comments below:
Introduction:
Minor note: The sentence in line 62-63 seems a bit hard to understand („During extreme weather conditions, such as severe heat waves, emergency medical services face increased demand and management challenges“) – maybe reformulate? E.g. „During extreme weather conditions, such as during severe heat waves, the demand for medical emergency services increases, leading to management challenges“?
Line 67, before quote 18: Space is missing (minor)
Line 80-81: Better change „drugs“ to „substance abuse“ or similar in order to distinguish it from prescribed medication (Original sentence: „This review aims to examine the biomedical literature to understand the connections between drugs and climate change“)
Methods:
Inclusion criterion: „Articles taken from the citations of the studies examined, were not bound by the
timeline of 5 years.“ – the way I understand it from your answers to my review-remark, you wanted to focus on recent results regarding original research, but where it seemed advisable for a deeper understanding of the topic, you also used older works from citiations that were not bound to original works either? Please mention clearly, also for readers.
Exclusion critieron: „Articles that do not relate to drug abuse correlate to climate change.“ – not happy with the formulation as the two-sidedness does not really become clear, maybe reformulate „articles without a (directional or non-directional) linkage between drug abuse and climate change“?
Exclusion criterion: Suggestion: To avoid confusions later with all the psychiatric medication studies that you later found, maybe mention explicitly at the „psychiatric-patient-exclusion-criterion“ that you did NOT exclude studies dealing with psychiatric medications as long as the participants were not clearly categorized as psychiatric patients.
Results:
Line 207: I can not access your supplement material, but if the „additional information“ included effect sizes, please mention explicitly, so readers know what awaits you in the supplement file.
Figure: Still displayed wrongly in my pdf-version… Should be in the results-part, not the discussion.
Discussion:
Thank you for considering most of my remarks and thank you for putting your discussion into a broader context.
The problem with psychiatric patients vs people taking psychiatric medication still is a little confusing when reading the paper (e.g. lines 260+). (As relatively extensive parts of your paper deal with these groups.) Maybe some introducing sentences could help, e.g. „even though not searched for them, a lot of studies deal with heat effects on psychiatric medication“. But maybe it becomes also a bit clearer with my suggestion regarding that in the „methods“-part.
Table 1: In my version this table is still displayed in the discussion section instead of in the results-section?
Line 269: Space is missing after quote nr. 42.
Overall, it still is a bit confusing to read so much about mental illnesses when psychiatric patients were explicitly excluded – though this part is very informative and should remain. Hence I suggest to introduce this part addressing explicitly to this problem, so that readers are not confused. E.g. something similar like „Even though studies with psychiatric patients have explicitly been excluded from our research strategies, overall results still pointed out repeatedly that people with mental health issues represent an especially vulnerable group in this context. [And then report whatever you have written here.]“
Line 277 typo: There is a „.“ too much.
Thank you for adding explicit limitations to the discussion.
Overall, I think the discussion improved a lot during the revision. […]”
We thank the reviewer for their detailed revisions, which have allowed us to improve the manuscript. Below is our response to the reviewer.
- “[…] Expressions are not always consistent (e.g. „substance addiction“, „drug abuse“, „substance abuse“, „substance misuse“etc.) – Please decide for a term […]”
Reply: We thank the reviewer for their observation. As per the reviewer's suggestion, we have adjusted the terminology to avoid any ambiguity. The changes have been highlighted in red.
- “[…] Introduction:
Minor note: The sentence in line 62-63 seems a bit hard to understand („During extreme weather conditions, such as severe heat waves, emergency medical services face increased demand and management challenges“) – maybe reformulate? E.g. „During extreme weather conditions, such as during severe heat waves, the demand for medical emergency services increases, leading to management challenges“? […]”
Reply: We thank the reviewer for their precise observation and suggestion. We have implemented the replacement, considering the reviewer's suggested phrase to be better.
- “[…] Line 67, before quote 18: Space is missing (minor) […]”
Reply: We thank the reviewer for their observation. The modification has been made as requested, and the space has been inserted.
- “[…] Line 80-81: Better change „drugs“ to „substance abuse“ or similar in order to distinguish it from prescribed medication (Original sentence: „This review aims to examine the biomedical literature to understand the connections between drugs and climate change“) […]”
Reply: We thank the reviewer for the observation. The modification has been made as requested.
- “[…] Methods:
Inclusion criterion: „Articles taken from the citations of the studies examined, were not bound by the timeline of 5 years.“ – the way I understand it from your answers to my review-remark, you wanted to focus on recent results regarding original research, but where it seemed advisable for a deeper understanding of the topic, you also used older works from citiations that were not bound to original works either? Please mention clearly, also for readers.
Reply: We thank the reviewer for their observation. The addition to the text has been executed exactly as requested.
- “[…] Exclusion critieron: „Articles that do not relate to drug abuse correlate to climate change.“ – not happy with the formulation as the two-sidedness does not really become clear, maybe reformulate „articles without a (directional or non-directional) linkage between drug abuse and climate change“? […]”
Reply: We thank the reviewer for their prompt observation. The indicated modification has been included as requested.
- “[…] Exclusion criterion: Suggestion: To avoid confusions later with all the psychiatric medication studies that you later found, maybe mention explicitly at the „psychiatric-patient-exclusion-criterion“ that you did NOT exclude studies dealing with psychiatric medications as long as the participants were not clearly categorized as psychiatric patients […]”
Reply: We thank the reviewer for their clarification. As requested, among the exclusion criteria, we have included reference to psychiatric patients.
- “[…] Results:
Line 207: I can not access your supplement material, but if the „additional information“ included effect sizes, please mention explicitly, so readers know what awaits you in the supplement file. […]”
Reply: We thank the reviewer for their observation. As requested by the reviewer, this additional clarification regarding the table has been inserted (highlighted in red in the text).
- “[…] Figure: Still displayed wrongly in my pdf-version… Should be in the results-part, not the discussion. […]”
Reply: We thank the reviewer for their observation. It is noted on this point that Table 1 is referenced in the Results section, as indicated by the reviewer. However, due to formatting constraints, we were unable to move it directly below that section. We will request MDPI to improve the text formatting accordingly.
- “[…] Discussion:
Thank you for considering most of my remarks and thank you for putting your discussion into a broader context.
The problem with psychiatric patients vs people taking psychiatric medication still is a little confusing when reading the paper (e.g. lines 260+). (As relatively extensive parts of your paper deal with these groups.) Maybe some introducing sentences could help, e.g. „even though not searched for them, a lot of studies deal with heat effects on psychiatric medication“. But maybe it becomes also a bit clearer with my suggestion regarding that in the „methods“-part. […]”
Reply: We thank the reviewer for their valuable observation. As requested, the modification has been accepted and highlighted in red within the text.
- “[…] Table 1: In my version this table is still displayed in the discussion section instead of in the results-section? […]”
Reply: I apologize for the inconvenience. Unfortunately, I am unsure how to resolve this issue, which evidently stems from a formatting transition. I will communicate this matter to MDPI.
- […] Line 269: Space is missing after quote nr. 42 […]”
Reply: We thank the reviewer for the observation. The modification has been carried out as requested.
- “[…] Overall, it still is a bit confusing to read so much about mental illnesses when psychiatric patients were explicitly excluded – though this part is very informative and should remain. Hence I suggest to introduce this part addressing explicitly to this problem, so that readers are not confused. E.g. something similar like „Even though studies with psychiatric patients have explicitly been excluded from our research strategies, overall results still pointed out repeatedly that people with mental health issues represent an especially vulnerable group in this context. [And then report whatever you have written here.]“ […]”
Reply: We greatly appreciate the reviewer for their observation and for their suggestion, which allows us to significantly improve the text. The text to be integrated has been inserted in red as requested.
- “[…] Line 277 typo: There is a „.“ too much. […]”
Reply: We thank the reviewer for the observation. The modification has been carried out as requested.
- “[…] Thank you for adding explicit limitations to the discussion […]”
Reply: We once again thank the reviewer. We are pleased that the changes have been approved by them.
Best Regards,
The corresponding author
